# Comparison between Traditional and Novel NMR Methods for the Analysis of Sicilian Monovarietal Extra Virgin Olive Oils: Metabolic Profile Is Influenced by Micro-Pedoclimatic Zones

**DOI:** 10.3390/molecules29194532

**Published:** 2024-09-24

**Authors:** Archimede Rotondo, Giovanni Bartolomeo, Irene Maria Spanò, Giovanna Loredana La Torre, Giuseppe Pellicane, Maria Giovanna Molinu, Nicola Culeddu

**Affiliations:** 1Department of Biomedical and Dental Sciences and Morpho-Functional Imaging (BIOMORF), University of Messina, Polo Universitario Annunziata, Viale Annunziata, 98168 Messina, Italy; arotondo@unime.it (A.R.); gbartolomeo@unime.it (G.B.); gpellicane@unime.it (G.P.); 2Department of Chemical, Biological, Pharmaceutical and Environmental Sciences, University of Messina, V.le F. Stagno D’Alcontres 31, 98166 Messina, Italy; irene.spano@studenti.unime.it; 3CNR—Istituto di Scienze delle Produzioni Alimentari (ISPA), Traversa La Crucca 3, Loc. Baldinca, Li Punti, 07040 Sassari, Italy; mariagiovanna.molinu@cnr.it; 4CNR—Istituto di Chimica Biomolecolare (ICB), Traversa La Crucca 3, Loc. Baldinca, Li Punti, 07040 Sassari, Italy; nicola.culeddu@icb.cnr.it

**Keywords:** NMR analysis, MARA-NMR, extra virgin olive oil, *Olea europaea*, ^13^C-NMR, metabolic profile, PLS-DA

## Abstract

Nuclear magnetic resonance (NMR) metabolomic analysis was applied to investigate the differences within nineteen Sicilian *Nocellara del Belice* monovarietal extra virgin olive oils (EVOOs), grown in two zones that are different in altitude and soil composition. Several classes of endogenous olive oil metabolites were quantified through a nuclear magnetic resonance (NMR) three-experiment protocol coupled with a yet-developed data-processing called MARA-NMR (Multiple Assignment Recovered Analysis by Nuclear Magnetic Resonance). This method, taking around one-hour of experimental time per sample, faces the possible quantification of different class of compounds at different concentration ranges, which would require at least three alternative traditional methods. NMR results were compared with the data of traditional analytical methods to quantify free fatty acidity (FFA), fatty acid methyl esters (FAMEs), and total phenol content. The presented NMR methodology is compared with traditional analytical practices, and its consistency is also tested through slightly different data treatment. Despite the rich literature about the NMR of EVOOs, the paper points out that there are still several advances potentially improving this general analysis and overcoming the other cumbersome and multi-device analytical strategies. Monovarietal EVOO’s composition is mainly affected by pedoclimatic conditions, in turn relying upon the nutritional properties, quality, and authenticity. Data collection, analysis, and statistical processing are discussed, touching on the important issues related to the climate changes in Sicily and to the specific influence of pedoclimatic conditions.

## 1. Introduction

Extra virgin olive oil (EVOO) is the hydrophobic fraction separated by milling drupes of *Olea europaea*. This fat is widely spread worldwide because of the perfect balance among saturated, mono-unsaturated, and poly-unsaturated fatty acids (SFAs, MUFAs, and PUFAs, respectively) [1]. The main benefits of EVOO diets are partially related to the high content of the MUFAs with respect to the other vegetable oils, because other minor species are certainly involved in key physiological pathways leading to biological effects. Among these, we note aliphatic and triterpene alcohols, hydrocarbons, volatile compounds, squalene (SQ), sterols, and the list is still open [2]. Among these, liposoluble phenols are very fashionable because of their demonstrated protective action against oxidative stress in vivo and in vitro [3,4,5]. The same chemicals work as EVOO’s self-protectors by stretching the famous shelf-life. Finally, phenolic species drive the organoleptic features (aroma and flavour, especially bitterness) of EVOOs, despite their relatively tiny molecular ratio [6,7].

Several analytical techniques have been employed to characterize EVOOs, mainly adopting separation devices coupled with very sensitive detectors. The EU commission still considers some of these protocols to be the official characterization procedures [8]. Namely, gas chromatographic separation and detection though the flame ionization (GC-FID) is chosen for the quantification of the fatty acids, which are present as glyceryl esters [9]; whereas many triacyl-glycerols (triglycerides) are detected and quantified through cumbersome high-performance liquid chromatography (HPLC), often coupled with the diode array detector (DAD), and the coupled methylated fatty acid residues are adopted to detect the fatty acid residues [10]. For the minor components such as terpenes and sterols, other GC-FID conditions with different derivatizations should be adopted, whereas the minor phenolic species would require another acetonitrile extraction and a different HPLC-DAD run [11]. Therefore, to detect the great variety of chemicals in EVOO, many target analytical methodologies have been proposed for the identification and quantification of specific compounds [12]. This explains the rapid growth of nuclear magnetic resonance (NMR) studies embracing the holistic approach to characterizing the chemical profile of complex mixtures [13]. Despite the claimed low sensitivity, the undeniable quick detection of all the organic species within matrices, without troublesome chemical treatments or separation procedures [14,15,16,17], explains the great success of this method for food analysis. We add that the pure quantitative spectral response and direct sampling minimize the many sources of errors, experimental time, and employment of reference standards [18].

Provided that the main challenge is represented by the relative sensitivity limits (respective to a specific compound) [19], unlike the traditional separation methods, NMR spectroscopy simultaneously detects all the organic molecules in the same sample, which are distinguished by their physical–chemical properties. In the last decades, many studies successfully demonstrated the wide application of NMR analysis of vegetable oils and in particular EVOOs [20] even though these were mainly employed to distinguish and cluster samples through multivariate statistical analyses [21], based on different features (cultivar, origin, climatic conditions, etc.) [22,23].

This study explores the ambitious idea to develop an NMR analytical protocol, based on the MARA-NMR algorithm [24], to reach an objective and absolute quantification by NMR of many different chemical species and verify that the NMR quantification is consistent with traditional methods. NMR metabolomic analysis is applied to investigate the differences within *Nocellara del Belice* monovarietal EVOOs sampled in two zones that are different in altitude and soil composition.

## 2. Results

### 2.1. NMR Metabolic Profile

The whole NMR data processing led to the quantification of metabolites reported in Table 1. The extended table referred to all the analyzed samples is reported in Table A1, and stacked spectra are also reported (Appendix A and Figure A1, Figure A2 and Figure A3). To complete the experimental panel, Table A2 displays all the outcomes retrieved by the traditional laboratory procedures. Specific measurements over three replicates for single samples revealed that the deviation on the output values was never over the 6% for the main metabolites and below the 12% for sterols and phenols. Several exchanges with experiments performed with a different number of scans led to similar values by the MARA-NMR algorithm [24] which further warrantees the general consistency of the method. Beyond the absolute quantification of metabolites in EVOOs, this study is addressed to the detection of the whole metabolic profile, which is demonstrated to change according to different environmental conditions. Eliminating some variables could help to reduce collinearity, leading to a more stable and reliable model. Indeed, after some trials, we have selected sixteen variables, which give a reliable and robust model [25].

### 2.2. GC vs. NMR Comparison

The NMR results are proportional to the molecular relative ratio, whereas traditional techniques usually are set to provide quantification in weight ratio respect to the total matter. This is why we have used the most reasonable conversion parameters in the comparative analyses.

This is leading to the necessary data conversion, which is in principle related to the molecular weight ratio. In the case of fatty acid quantification by GC-FID, this subject is further complicated by the fact that the real main components in EVOOs are fatty esters [26], owning a totally different molecular weight. This justifies our idea to keep the ratio among fatty esters in “molecular ratio”, and therefore the other minor components are first scaled as molecular quantities. Afterward, it is possible to convert these back to weight ratios. Provided that the GC and NMR techniques provide basically different information [26], Figure 1 represents the compared quantification according to our absolute independent scaling factors. The fitting comparison is astonishingly good, and usually GC-NMR is overestimating SFA (saturated fatty acids) and TUFA (three-unsaturated fatty acids) and underestimating MUFA (mono-unsaturated fatty acids) and DUFA (di-unsaturated fatty acids). This is anyway within the standard deviation and not always occurring.

### 2.3. Total Phenolic Compounds: NMR vs. Folin-Ciocolteau

We point out again that the NMR direct quantification of the four most representative phenolic compounds (Oleocanthal, Olaceine, Oleuropein, and Ligstroside aglycones) is in molecular ratios [27], just as is the spectrophotometric response, which, though, is usually reported in gallic acid equivalents and anyway is considered the whole class of compound. This is why our reasonable idea was to compare results, keeping in mind the gallic acid molecular weight as a basic “data converter” (Figure 2). It appears that the popular, easy, and direct spectrophotometric method is affected by the use of a three-functional hydroxy group chemical reference (gallic acid) and by the interferences of inhibitory, additive, and enhancing types [28]. Provided that the general trend is well respected, the few discrepancies are reasonably due to the different number of functional groups per molecules and to the possible interferences in the Folin–Ciocâlteu measurements [29]. We would like to mention that the NMR method quantifies a fifth elenolic derivative called Elenolide [30], which is not considered an electron-donating compound, and it is not considered in this comparison. It should be noted that samples N_7, N_8 and N_12 with remarkable presence of TPC are also those displaying the higher deviation with an underestimate of the spectrophotometric method. We might argue that it could be due to the inhibitory interference of the Folin reagent within the experimental times.

In the last five years, Sicily has faced an unprecedented drought, with an average rise of temperatures that could theoretically lead to a dramatic drop of the phenolic composition. Provided that our NMR analysis is consistent with other experimental evidence in this paper, as well as in others [27], the phenolic fraction records a general slight decrease by keeping its important feature in the *Nocellara del Belice* oils. We think that a moderate thermal stress of plants is even better, promoting the production of phenolic species despite their thermal stability, which indeed decreases the phenolic presence after drastic oil processing [31]. Differences in mean temperatures lead to differences in oleic, linoleic and others lipid acids [32].

### 2.4. Statistical Analysis of the Metabolic Profile

Statistical graphs are reported in Figure 3. For the statistical analysis, we have used 19 observables and 16 variables. The PCA (Principal Component Analysis) shows a promising clustering of samples according to specific variables (loadings not shown), revealing a fair distribution that may be linked to several factors (i.e., temperature, altitude, solar exposition and/or pedologic). The Cumulative Sum of Squares R^2^ 0.606 and the fraction of the total variation of X Q^2^ 0.176 indicate an insufficient fit, which may depend on the low number of samples. The PCA shows clustering into two groups, L1 and L2 (Figure 3a).

Using unbiased methods like PCA provides a first confirmation before moving on to supervised methods. Ideally, the outcomes of PCA analyses help us to form an initial hypothesis, which can then be further tested and verified using OPLS-DA [33].

The PLS-DA (R^2^ 0.889 and Q^2^ 0.789) shows a reliable separation between L1 and L2 samples (Figure 3b). The robustness of the model was tested by random permutation and leave-one-out tests, both yielding 100% robustness.

In PLS-DA analysis, a permutation test evaluates the significance of the model by checking whether its quality parameters remain consistent when the dataset rows are shuffled, influencing the reliability of the conclusions. In order to rule out that these results were due to overfitting, a 100-fold permutation test was performed. The resulting regression lines showed an R2 intercept at 0.325 and a Q2 intercept at −0.3131, indicating a valid model. These tests indicate that the model is robust and useful for future dataset extensions.

The misclassification test (Table 2) summarizes the number of observations with known class memberships that were correctly classified in class or PLS-DA models. All samples are correctly classified.

From Figure 4, we can identify the most important variables and their roles in the PLS-DA mode. These are linoleate, linolenate, palmitic, and *cis*-vaccenic esters for the L2 group (negative contribution), and oleate and polyphenols for the L1 group (positive contribution). These results are consistent with those obtained by Piravi-Vanak et al. (2012) [34], which demonstrated that the fatty acid composition of olive oil is strongly influenced by the variety, ripening process, and geographical origin, particularly latitude and climatic conditions.

## 3. Discussion

The variables affecting olive oil quality before the production process include the variety (genetic factor), tree age, cultivation area/geographical origin (latitude, longitude, and elevation), climate (temperature, rainfall, humidity, and wind speed), and soil texture and composition [35]. Samples of monovarietal EVOO from a limited sampling area reduce the field of variability to altitude/temperatures and soil differences and references therein.

Our results demonstrate that 1H-NMR-based analysis enables the detection of significant metabolic differences in monovarietal oils coming from the same “Belice valley” (western Sicily), making these results consistent with other traditional analytical measurements. The detected variabilities are obviously influenced by the terroir condition, and despite the limited gamma of independent samples, we demonstrate that the metabolomic profile is strictly related to the specific microclimatic territory.

The monovarietal plants are fixing the genetic features, and therefore the detected differences mostly refer to terroir and to some kind of specific agronomic practice. Variations in FA composition during ripening are closely associated with genetic factors and seasonal weather conditions [36], determining positive and negative trends in oleic, palmitic, and linoleic acids concentrations. Looking at the loadings, the oleic acid is positively linked to samples coming from the seaside plains and is dominating, the main component being the linoleic acid related in an inverse mode. So, this dimension is strictly discriminating the altitude of the crops, which is basically the main discriminant factor for a monovarietal species. The second dimension is mainly made by saturated fatty acids, and it is known that squalene, linoleic, palmitic and oleic acids are mainly influenced by the thermal regime of the summer period, increasing linoleic, squalene and palmitic acid as temperature rises, while opposite the effect is observed on oleic acid.

The area of *Nocellara del Belice* features varying altitudes that influence local agriculture. Lower altitudes tend to have milder temperatures and different soil compositions, which affect crop types and yields. Higher elevations experience cooler temperatures and potentially more wind, impacting the growing conditions for certain plants. These altitude differences may contribute to the diversity of the FA profile in EVOO.

The general trend of our data well matches Ivavov’s role. Indeed, the L1 group, cropped at higher altitude, showed a lower mean oleic composition (O, 62.8 ± 0.6%) and higher mean linoleic composition (L, 10.9 ± 0.5%). The fatty acid composition of the samples collected from the L2 group suggests that EVOOs belong to the high oleic–low linoleic group (mean oleic acid 66 ± 1 and mean linoleic acid 8.7 ± 0.7).

These preliminary encouraging results have shown a minimal influence of the soil on the chemical–physical and thermal properties of EVOOs. However, further analysis of a larger set of samples would be needed to confirm these findings.

## 4. Materials and Methods

### 4.1. Samples

The nineteen chosen samples were selected according to the orchard cultivation techniques, whose production amounts could guarantee independent olive milling. Oil mill plants are “Oleificio Asaro” (L1, six samples) and Campobello di Mazzara “Keolive” (L2, thirteen samples). The L1 and L2 sampling regions featured the pedoclimatic conditions reported in Table 3.

All the samples were guaranteed to be the monovarietal Sicilian cultivar *Nocellara del Belice*, one of the most diffused cultivars in the region. The samples were analyzed in triplicate directly after production. Each sample was initially stored in capped glass tubes at room temperature inside a dark and dry cupboard, then preserved at −40 °C until analysis to promote the long-term stability of the phenols. Before analysis, each sample was defrosted at room temperature in a dark room.

Figure 5 indicates the rough sampling areas. Part (a) shows differences in altitude between zone L1 (red), which is meanly 250–300 mt higher than L2 (blue). Part (b) shows differences in soil composition. Zone L1 (red) is a mix of Brown Soils and Lithosols and Vertosols. Zone L2 (blue) is a mix of red regosols, regosols on clay rocks, and alluvial soils. The peculiar territory in Figure 1 offers different elements, which are challenging a neat selection of environmental influences on the composition of olive oil. Since the homogeneity of the harvesting periods and extraction conditions allowed us to analyze the variability of the fatty acid (FA), this is potentially affected by altitude, solar exposure, and pedological conditions.

### 4.2. Chemicals

Tetramethyl silane (TMS), high-purity deuterated chloroform (CDCl_3_), methanol, *n*-hexane, cyclohexane, and chemicals at reagent grades were supplied from Sigma-Aldrich (Milan, Italy).

### 4.3. NMR Sample Preparation

Sample preparation was developed after several test experiments. The experimental setup is our best optimization to keep a reasonable field homogeneity affecting line-width and resolution, sensitivity, and chemical stability. For instance, we would like to point out that many studies suggested adding small amounts of d6-DMSO to decrease the sample viscosity and therefore the field homogeneity, and this is supported by several quantification studies in polar media [37,38] as well as in CDCl_3_ medium (for vegetable oils or extracts) [28,39]. On the other hand, our experiments demonstrate that, in the case of secoiridoid derivatives, polar media trigger isomerization and decarboxylation processes, preventing specific quantification, as shown in Figure A4. We point out that, unlike the mentioned studies, this quantification is based on the CHO aldehydic group, whose signal is not affected by the potential hydrogen bonds of the polar media. Another possible strategy was to use small amounts of EVOOs (20 μL), gaining the great advantage of smooth homogeneity and definite spectral lines [40]. However, this technique is advisable for detecting only the main components of protonic experiments, whereas it would increase the experimental time for the detection of less sensitive signals like ^13^C resonances or ^1^H resonances related to minor components.

All the oil samples were dissolved in CDCl_3_. In some test cases, we used traces of TMS as a reference standard, and later we avoided this “contamination”, since other EVOO signals can be used instead as references for the NMR analysis. As suggested elsewhere [18], we kept the oil-to-CDCl_3_ weight ratio equal to 13.5:86.5; it corresponds to a mixture of 122 μL of oil and 478 μL of deuterated chloroform into a 5-mm test tube for NMR. Tubes were immediately sealed to prevent the solvent evaporation which might affect the chemical shift of many signals, especially the olefinic and carbonylic ^13^C signals; the constant concentration is a crucial experimental task to be performed, and it will become the main source of non-systematic errors as we run the phenol quantification in absolute measurement mode.

### 4.4. NMR Experimental Protocol

For any sample, three basic experiments were recorded:Experiment A: a standard protonic spectrum with 16 scans and a suitable cycling delay for quantitative analysis.Experiment B: ^1^H- DPFGSE (double-pulsed gradient spin echo) spectrum [27] with 32 scans for the detection and quantification of aldehydic phenolic species.Experiment C: full-time ^1^H decoupled ^13^C spectrum with 32 scans with a suitable recycling delay for quantitative evaluations [24].

The experimental choices owe to our personal optimization, seeking high precision and fast acquisition, and certainly these could be differently tuned and maybe improved. For instance, the ^1^H experiment could be drastically shortened according to Castejon et al. 2014 [41] (just 4 scans), if we should not consider the sterolic and phenolic fractions. Our three experiments’ setup lasted around 9 min, 9 min, and 35 min, respectively; therefore, by including the preparation procedure, the total experimental analysis for every sample was 60 min in the worst-case scenario.

### 4.5. NMR Acquisition and Processing

Samples in the 5 mm NMR tubes were analyzed by a 500-MHz spectrometer (Agilent Technologies, Milan, Italy) equipped with a new generation probe with gradients (ONE_nmr probe) at the constant temperature of 298 K. After the automatic tuning and gradient shimming, the line shape of the TMS signal was checked by shimming until the line shape was lower than 1.5 Hz. The 1D ^1^H and {^1^H}-^13^C NMR spectra were run at 499.74 and 125.73 MHz, respectively. For every experiment, we have chosen to use 90° pulses for maximum sensitivity coupled with recycling delay able to totally relax the saturation (more than five times T1). The hard pulse for the maximum sensitivity (90° pulse) was calibrated throughout the samples and was always within 8.2 ± 0.1 μs at 58 dB. ^1^H-NMR experiments (experiment A) were run with a spectral width of 12 ppm, with 16 scans, 12 s of acquisition delay, and 3 s of acquisition time, so that the 15 s recycle delay was enough to keep quantitative methods, regardless of the different protonic relaxation times (maximum value of T1 = 2.5, which is less than 5 times the total recycling time). For the same reason, some totally decoupled {^1^H}-^13^C spectra (experiment C) were first acquired with the 90° hard pulse (11.2 ± 0.3 us at 6 dB), 64 scans, 2 s of acquisition time, and 25 s for the time delay. Afterward, these experiments were compared to some others with lower recycling time and a smaller tilting angle, with the purpose of optimizing the experimental time, keeping the quantitative ratio of ^13^C signals. The final optimized conditions were: 96 scans, 85° pulse, and 18 s of total recycling delay, for a total of 35 min experimental time. According to our past studies, we also optimized the selection through selective pulses of the aldehydic region (8–10 ppm range) to perform a protonic DPFGSE spectrum (experiment B) [31,42] based on the standard ^1^H experiment (A); we harnessed the SEDUCE (SElective Decoupling Using Crafted Excitation) shaped pulse between 8 and 10 ppm with a duration of 100 ms [43]. After 8 to 16 scans, the spectrum displayed signals in the region of interest (barely foreseen through experiment A) with a better signal-to-noise ratio, provided that a slight line-broadening function (0.3 Hz) was applied for the Fourier-transform procedure.

Experiment A was frequency calibrated by using the internal β-sitosterol signal (δ_H_ = 0.738 ppm). The frequency of this signal is double-checked in several samples by using the external frequency of TMS (δ_H_ = 0.0), which never shifted the spectral profiles more than 0.005 ppm. Similarly, for ^13^C calibration (experiment C), the divinyl methylene group of linoleate glyceryl esters (L11; δ^13^C= 25.6614 ppm) was used, always keeping the known TMS ^13^C signal to δ^13^C = 0.0 ± 0.05 ppm. Provided that it is here demonstrated that the TMS calibration would not really change results, the calibration over internal signals is preferred, respective to the TMS calibration, because (a) this could be further used and extended for samples without the internal reference, and (b) TMS is a very small isotropic molecule with very long longitudinal relaxation times (T1_1H_ > 5s and T1_13C_ > 12 s as measured by inversion recovery experiments).

### 4.6. NMR Processing Strategies and Quantification

The basic principle of the NMR quantification (qNMR) is that any signal is strictly proportional to the relative representation of that chemical group embedded in its parent molecule [44]. Of course, the number of magnetically equivalent nuclei is to be considered as a scaling factor [24], and possible overlaps should be accounted for in the general outcome [45]. According to this scientific background, the NMR power is related to the chance to assess the relative concentration of several molecules, even just focusing on a specific spectral region of an NMR experiment. The idea to perform three different spectra is pursued to collect a great number of data, all related by the same mentioned quantification roles. Briefly, the MARA-NMR method [24] is a best-fitting regression of hundreds of qNMR relationships, aimed to sort out the best-fitting quantification model for nineteen independent metabolites. The low quadratic deviations (rho) found for any sample (observable) warrantees the self-consistency and robustness of the final outcomes. Practically speaking, the three mentioned experiments, A, B and C, were processed, baseline-corrected, and finely aligned. Finally, the experiments were integrated in 102, 18, and 97 chosen bins. The final matrix was made by 19 observables and 220 spectroscopical variables. The used quantification equations were 13, 7, and 70, involving integrals of the three experiments, respectively, inferring the quantification of twenty-four independent compounds (among these, just sixteen presented significative variances not affected by shelf-life and thus relevant for our studies). In this study, we harnessed different processing procedures: (1) reload spectra in MestreNova (MestreNova 6.6.2) running on a Windows 10 laptop, trying multiple spectra treatments with calibration, automatic phasing, alignment, and group-integration to retrieve the general matrix; (2) spectra loading in Topspin 4.2.0, serial treatment for calibration, phasing, base-line correction, and multiple integration through an au-program, so that, again, we could retrieve the data-matrix for the three experiments; (3) trying the innovative built-in icoshift [37] within the Matlab R2018a software package, customized to match the three model-experiments. The last MATLAB procedure issued the lowest rho (deviation coefficient) and displayed the best consistency, probably because of the best local alignment allowed by this routine.

### 4.7. Traditional Analytical Essays

To assess that the analyzed oils fall in the range of EVOO free fatty acidity (FFA), peroxide value (PV) and specific spectrophotometric indices were determined.

The FFA of each EVOO sample was estimated by titration, according to [8], as follows: 5 g of oil were mixed with 90 mL of ethyl alcohol/diethyl ether (1:2, *v/v*), added to the pH indicator (phenolphthalein), and the resulting solution was titrated with 0.1 M NaOH. The results were expressed as a percentage of oleic acid.

The PV value was determined from a solution of 1 g of oil mixed with 25 mL of acetic acid/chloroform (3:2, *v*/*v*), added to 0.5 mL of a KI-saturated solution. The resulting mixture was put in the dark for 5 min; subsequently, 75 mL of distilled water were added, and the mixture was titrated with 0.01 N Na_2_S_2_O_3_, using starch paste as an indicator. The results were expressed as meq of O_2_/kg oil.

The determinations of the spectrophotometric indices were carried out by dissolving 0.1 g of oil with pure cyclohexane in a 10 mL volumetric flask. Each sample was analyzed using an UV/Vis spectrophotometer Shimadzu, Model UV-2401PC (Shimadzu Italia s.r.l., Milan, Italy). Specific UV absorbance at 232 and 270 nm (K_232_ and K_270_, respectively) and ΔK were determined. The values were determined following the analytical methods described in [8], to attribute the oil samples to a commercial class.

### 4.8. Gas-Chromatographic (GC) Analysis of Fatty Acid Methyl Esters (FAMEs)

The fatty acid composition of the olive oil samples was carried out by GC-FID after derivatization to their methyl esters (FAMEs) [46]. A gas chromatograph (GC) (Master GC-DANI, Milan, Italy), equipped with a split/splitless injector, a flame ionization detector (FID), and a capillary column (Phenomenex ZB-Wax, 30 m × 0.25 mm, film thickness 0.25 µm), was used for the GC analysis. The following chromatographic conditions were used: the oven temperature was programmed from 50 °C (2 min) to 210 °C (at 3 °C min^−1^) and then set to be isothermal for 15 min; the injector and detector temperatures were 240 °C; a constant linear velocity of ultrapure helium (carrier gas) was used at 30 cm s^−1^. The injection volume was 1 µL, with a split ratio of 1:50. The identification was carried out by comparing the retention times of the compounds identified in the oil to the retention times of a reference FAME mixture. The percentage of individual FAMEs was calculated relative to the total area of the chromatograms. All determinations were performed in triplicate. Data processing was carried out using *Clarity Chromatography v.4.0.2* software.

### 4.9. Quantification of Total Phenol Content (TPC)

The TPC of each oil sample was determined by the Folin–Ciocâlteu spectrophotometric assay, referring to the Dordevic method [47] with slight modifications. Briefly: 6 mL of oil were added to 6 mL of a H_2_O/MeOH (80:20, *v*/*v*) solution, mixed, and centrifuged. Then, 0.2 μL of supernatant were added to 1.8 mL of distilled water, 8 mL of Na_2_CO_3_ solution (75 g/L), and 10 mL of the Folin–Ciocâlteu reagent, diluted 1:9. After about 2 h at 20 °C, the TPCs were measured at 760 nm using a UV spectrophotometer (UV-2401 PC, Shimadzu, Kyoto, Japan). The results were expressed as gallic acid equivalent (GAE, mg kg^−1^)

### 4.10. Statistical Analysis

Statistical analyses were performed using SIMCA-P software version 13.0 (Umetrics AB, Umea, Sweden) for statistical analysis. Data were scaled using the UV function in SIMCA-P. PCA (Principal Component Analysis) data analysis was performed for exploratory purposes and outliers’ recognition, while Projection to Latent Structures (PLS)-based methods were used for discriminant analysis and data set comparison. We used an Orthogonal extension of PLS-DA [25], in which the first latent variable accounted only for correlated data variations. PLS-DA models were evaluated using the goodness-of-fit parameter (R2Y) and the predictive ability parameter (Q2Y). First, 19 independent samples were chosen to run a multivariate statistical analysis without any defined classification. Because of the relatively limited available samples, we preferred to use the sixteen variables issued by the NMR quantification to avoid overfitting, collinearity, and challenging recovery of the chemical rationale.

To check whether there were any differences in variables to classify samples and to create predictive models, we used PLS-DA (Discriminant Analysis) analysis (Simca-P v 12, Sartorius Gottingen—Germany). Consequently, we tried to correlate the compositions of EVOO with discrete variables such as the place of origin, mean rainfall per year, or altitude of orchards. The results were analyzed, and the robustness of the results was verified using the methods proposed by Trygg [48]. A permutation test was performed, randomly changing the class attributions of 3 samples for time.

## 5. Conclusions

In this paper, we demonstrate that the novel NMR analysis and MARA-NMR processing enable the direct quantification of several components in EVOO in a straightforward way. This prompted us to customize a model suitable to discriminate homogeneous samples taken from slightly different crops. The explorative PCA was harnessed as a base for multi-variate analysis toward the predictive OPLS-DA model. Specifically, we have demonstrated that monovarietal EVOOs produced in the same region with similar solar exposition display a composition variability that is mainly affected by the altitude. The whole presented protocol reveals an unprecedented potential to run further EVOO studies aimed at the targeted investigation of factors (different cultivar, belonging, and agronomical practice) influencing the olive oil composition. This methodology proposes the construction of chemometric models for the division into production areas, which could be a key tool for the allocation of the EU Protected Designation of Origin “*Nocellara del Belice*” and for safeguarding this valuable product.

## Figures and Tables

**Figure 1 molecules-29-04532-f001:**
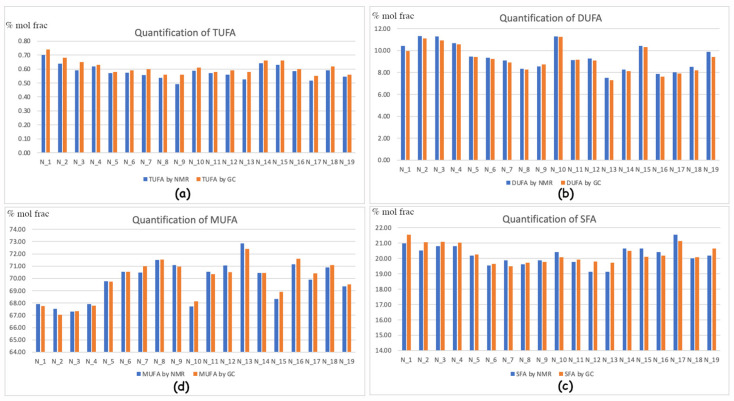
Histograms representing the compared quantification of the fatty acids/esters by NMR and GC processing in % of the molecular fraction: (**a**) TUFA, which is linolenic acid/esters (Ln); (**b**) DUFA linoleic acid/esters (L); (**c**) Total mono-unsaturated fatty esters (MUFA, which is O + V + PO); (**d**) Saturated fatty acid esters (SFA, which is mostly P + S and other minor fatty acids).

**Figure 2 molecules-29-04532-f002:**
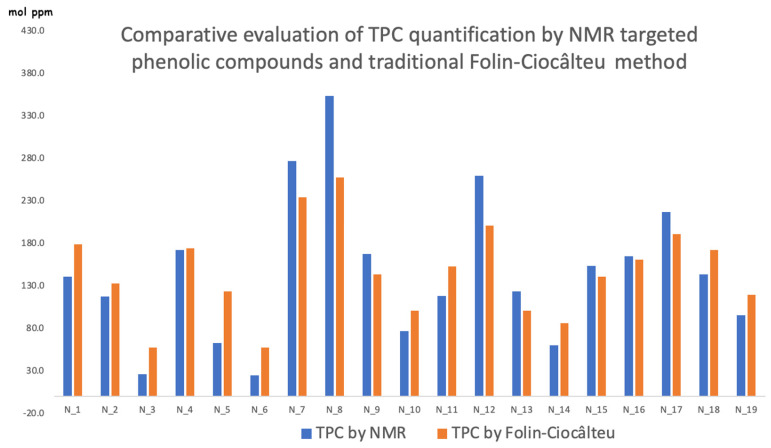
Comparison by histograms of the total phenolic content measured by NMR and through the traditional spectrophotometric Folin–Ciocâlteu method.

**Figure 3 molecules-29-04532-f003:**
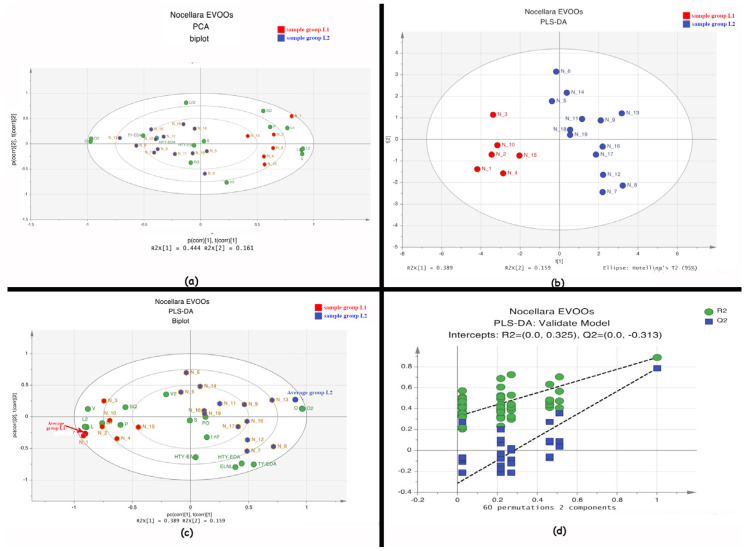
Statistical graphs concerning the multi-variate analyses: (**a**) PCA biplot showing two score groups (in red and blue) and loadings (green spots) R^2^ 0.606 Q^2^ 0.176; (**b**) PLS-DA score plot showing two L1 and L2 groups with R^2^ 0.889 and Q^2^ 0.789; (**c**) PLS-DA biplot; (**d**) 60-fold permutation.

**Figure 4 molecules-29-04532-f004:**
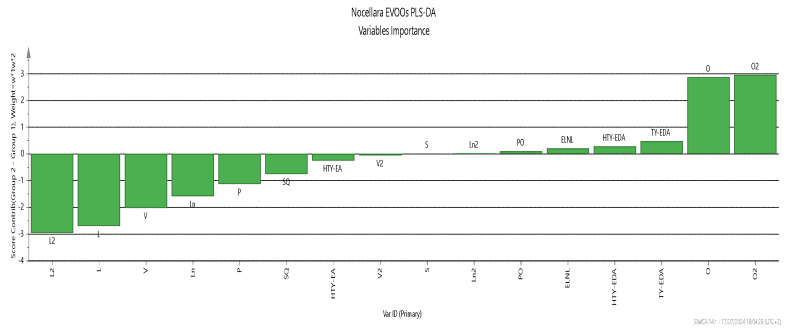
Score contribution that summarizes the importance of the X-variables for both the X- and Y-models.

**Figure 5 molecules-29-04532-f005:**
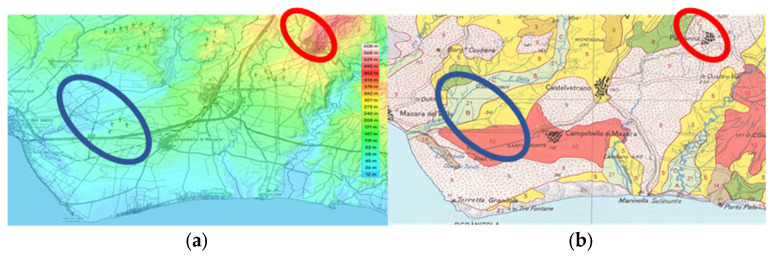
Maps reporting the main differences between sampling zones (**a**) Topography (**b**) Soil description and associations. Data were obtained from Sit-AGRO (Sicily Region- Sistema Informativo Territoriale—https://www.sitagro.it/jml/, accessed on 1 May 202).

**Table 1 molecules-29-04532-t001:** **NMR detected** and quantified metabolites, along with the relative label code used in the statistical methods.

Metabolites	Code
**Squalene molecular %**	**SQ**
**Linolenate esters %**	**Ln**
**Linoleates esters %**	**L**
**Oleic esters %**	**O**
**Palmitoleic esters %**	**PO**
**cis-vaccenic esters %**	**V**
**palmitate esters %**	**P**
**sterarate esters**	**S**
**Internal * Linolenate esters %**	**Ln2**
**Internal * Linoleates esters %**	**L2**
**Internal * Oleic esters %**	**O2**
**Internal * cis-vaccenic esters %**	**V2**
**Oleocanthal**	**TY-EDA**
**Olaceine**	**HTY-EDA**
Ligstroside aglycone (all the derivates)	TY-EA
**Oleuropein aglycone (all the derivates)**	**HTY-EA**
**Elenolide**	**ELNL**
total Phenolic species	TPH

* Internal refers to the 2- esterification point over the glycerol moiety of tri- and di-glycerides; the saturated esters are excluded because of their irrelevant presence. The variables in bold are those used for the final statistical model.

**Table 2 molecules-29-04532-t002:** The misclassification table (leave-one-out).

	Members	Correct	1	2	No Class (YPred ≤ 0)
**1**	6	100%	6	0	0
**2**	13	100%	0	13	0
**No class**	0		0	0	0
**Total**	19	100%	6	13	0
**Fisher’s prob.**	3.7 × 10^−5^				

**Table 3 molecules-29-04532-t003:** Data retrieved from Sit-AGRO—Sicily Region- Sistema Informativo Territoriale—URL https://www.sitagro.it/jml/ (accessed on 1 May 2024) reporting the pedoclimatic conditions of the sampling regions.

Zone	Partanna (L1)	Campobello di Mazzara (L2)
Altitude	350–400	70–120
T (average)	15/16 °C	18/19 °C
Soil	Brown Soils, Lithosols, Vertosols	Red Regosoils, Regosoils on Clay Rocks
Solar exposition	1651 kW/year	1655 kW/year

## Data Availability

Data sharing is not applicable.

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
