# Peer review of "Comparison between Traditional and Novel NMR Methods for the Analysis of Sicilian Monovarietal Extra Virgin Olive Oils: Metabolic Profile Is Influenced by Micro-Pedoclimatic Zones"

_molecules, 2024, doi:10.3390/molecules29194532_

Round 1

Reviewer 1 Report

Comments and Suggestions for Authors

 In this manuscript the authors performed a comparison between traditional and innovative NMR methods for the analysis of Sicilian monovarietal EVOOS. I have some considerations related to the paper.

1.       Please note that references 12 and 13 are the same, moreover in the lines 68-74, the authors should consider other relevant references focused on the EVOOS NMR metabolomic analysis, as, for example:

https://doi.org/10.3390/molecules28041738

https://doi.org/10.3390/molecules26082233

https://doi.org/10.1111/1541-4337.13005

2.       paragraph 2.1: this section should be improved since it is not clear. First of all it should be moved in  the experimental section. Moreover, a table reporting the differences between the two areas could be useful.

3.       Line 129:no gray back-ground is visible in table 1, so it is not possible to identify the sixteen selected variables

4.       line 178: the loading line plot for the pca model should be added in order to identify the variables responsible for the separation.

5.       line 178-179 . From the observation of the score plot, a clustering of the samples according to the producers (M1 and M2) a can be observed, but the authors stated "a fair distribution which may be linked to several factors (i.e., temperature, altitude, solar exposition and/or pedologic). Please explain the influence of these factors and report the temperature, altitude, solar exposition and/or pedologic data. It is not clear the influence of the sampling area on the distribution, if the oil samples provider M1 and M2 are from L1 and L2 areas paragraphs 2.1 (sampling areas) and  4.1 (samples) should be combined

6.       section 5 , conclusion: this paragraph should be improved. Please better described the findings and the innovation of your research

Minor comments:

Please define all the acronyms in the manuscript, LINE 26 what is MARA?

line 184 the sentence " If there are multiple panels, they should be listed as:" should be removed

line 450-452 the sentences "Please turn to the CRediT taxonomy for the term explanation. Authorship must be limited to those who have contributed substantially to the  work reported." should be removed

Author Response

Review 1

 In this manuscript the authors performed a comparison between traditional and innovative NMR methods for the analysis of Sicilian monovarietal EVOOS. I have some considerations related to the paper.

Comment 1.1:

Please note that references 12 and 13 are the same, moreover in the lines 68-74, the authors should consider other relevant references focused on the EVOOS NMR metabolomic analysis, as, for example:

https://doi.org/10.3390/molecules28041738

https://doi.org/10.3390/molecules26082233

https://doi.org/10.1111/1541-4337.13005

Reply 1.1

Thanking the reviewer for the shared good editing we have modified the manuscript accordingly: the references 12 and 13 has been corrected and the suggested references added.

Comment 1.2:

paragraph 2.1: this section should be improved since it is not clear. First of all it should be moved in  the experimental section. Moreover, a table reporting the differences between the two areas could be useful.

Reply 1.2:

The entire paragraph was moved to the material and methods, the wrong M notation is totally replaced with L and the two producers and production zones are detailed in the results section hoping to meet the reviewer suggestion and the better quality of the manuscript.

Comment 1.3

Line 129:no gray back-ground is visible in table 1, so it is not possible to identify the sixteen selected variables

Reply 1.3:

The variables used for the statistical analysis are now evidenced in bold and accordingly indicated in the footnote.

Comment 1.4 :

line 178: the loading line plot for the pca model should be added in order to identify the variables responsible for the separation.

Reply 1.4:

Figure 4, now become Figure 3, was modified according to the requests.

Comment 1.5:

line 178-179 . From the observation of the score plot, a clustering of the samples according to the producers (M1 and M2) a can be observed, but the authors stated "a fair distribution which may be linked to several factors (i.e., temperature, altitude, solar exposition and/or pedologic). Please explain the influence of these factors and report the temperature, altitude, solar exposition and/or pedologic data. It is not clear the influence of the sampling area on the distribution, if the oil samples provider M1 and M2 are from L1 and L2 areas paragraphs 2.1 (sampling areas) and  4.1 (samples) should be combined

Reply 1.5:

This point is of course related to the point 2. Thanks to this reviewer we have deeply modified several paragraphs (2.1, 4.1, the final part of the “Discussion” and Conclusions) trying to improve the clarity and the impact of this paper. In order to show the soil and climate differences, table 3 has been added to the materials and methods.

Comment 1.6: section 5 , conclusion: this paragraph should be improved. Please better described the findings and the innovation of your research

Reply 1.6:

As reported above the Conclusions were edited to our best.

Comment 1.7, Minor comments:

Please define all the acronyms in the manuscript, LINE 26 what is MARA?

Reply 1.7:

The acronym has been clarified.

comment 1.8:

line 184 the sentence " If there are multiple panels, they should be listed as:" should be removed

Reply 1.8:

The sentence was removed.

Comment 1.9:

line 450-452 the sentences "Please turn to the CRediT taxonomy for the term explanation. Authorship must be limited to those who have contributed substantially to the  work reported." should be removed

Reply 1.9:

The sentence was removed.

Again we are very grateful for the appropriate text revision of the reviewer

Reviewer 2 Report

Comments and Suggestions for Authors

The manuscript presents an interesting study comparing traditional and innovative Nuclear Magnetic Resonance (NMR) methods to analyze the metabolic profile of monovarietal Sicilian Extra Virgin Olive Oils (EVOO), focusing on the impact of micro-pedoclimatic conditions. The research addresses an important topic, especially given the growing interest in understanding how regional environmental factors influence the quality and composition of EVOOs. The work is scientifically relevant and provides valuable insights into the application of NMR for food analysis. However, there are several areas where the manuscript could benefit from significant revisions to improve clarity, depth, and overall presentation.

Major Points:

-While the NMR experimental protocols are described in detail, the justification for choosing specific parameters (e.g., pulse sequences, relaxation times) is not well-explained. The manuscript would benefit from a more comprehensive discussion on why these parameters were chosen and how they impact the results. Additionally, the potential limitations or challenges encountered during NMR data collection and processing should be discussed.

-The results section presents a wealth of data, but the presentation is sometimes unclear. The authors should consider reorganizing the data to enhance readability. For example, histograms and figures (e.g., Figure 2 and Figure 3) should be accompanied by more detailed explanations in the text, highlighting the most significant findings and their implications.

-The statistical analysis, including PCA and PLS-DA, is a key component of the study. However, the manuscript reports poor fit results for PCA and yet proceeds with PLS-DA. This approach needs more justification. The authors should explain why they proceeded with supervised analysis despite the poor PCA results and discuss how the limited sample size may have impacted the robustness of the models. Additionally, the permutation test results and their implications should be elaborated on.

Addressing the points raised in this review will strengthen the manuscript and make it more accessible and informative to readers.

Author Response

Review 2

The manuscript presents an interesting study comparing traditional and innovative Nuclear Magnetic Resonance (NMR) methods to analyze the metabolic profile of monovarietal Sicilian Extra Virgin Olive Oils (EVOO), focusing on the impact of micro-pedoclimatic conditions. The research addresses an important topic, especially given the growing interest in understanding how regional environmental factors influence the quality and composition of EVOOs. The work is scientifically relevant and provides valuable insights into the application of NMR for food analysis. However, there are several areas where the manuscript could benefit from significant revisions to improve clarity, depth, and overall presentation.

Major Points:

Comment 2.1:

-While the NMR experimental protocols are described in detail, the justification for choosing specific parameters (e.g., pulse sequences, relaxation times) is not well-explained. The manuscript would benefit from a more comprehensive discussion on why these parameters were chosen and how they impact the results. Additionally, the potential limitations or challenges encountered during NMR data collection and processing should be discussed.

Reply 2.1:

We are grateful to reviewer 2 for spending her/his time in the editing and improving process

We have edited paragraph 4.4 to fulfill the reviewer requests, NMR experimental protocols was been better detailed, the discussion improved and, the potential limitations or challenges discussed.

Comment 2.2:

The results section presents a wealth of data, but the presentation is sometimes unclear. The authors should consider reorganizing the data to enhance readability. For example, histograms and figures (e.g., Figure 2 and Figure 3) should be accompanied by more detailed explanations in the text, highlighting the most significant findings and their implications.

Reply 2.2:

We have tried our best to refine the presentation of the comparative data in the new paragraphs 2.2 and 2.3

Comment 2.3:

-The statistical analysis, including PCA and PLS-DA, is a key component of the study. However, the manuscript reports poor fit results for PCA and yet proceeds with PLS-DA. This approach needs more justification. The authors should explain why they proceeded with supervised analysis despite the poor PCA results and discuss how the limited sample size may have impacted the robustness of the models. Additionally, the permutation test results and their implications should be elaborated on. Addressing the points raised in this review will strengthen the manuscript and make it more accessible and informative to readers.

Reply 2.3:

We are grateful for the clever analytical observation, we have modified lines 202-220. We would like to point out that we have used PCA exclusively as a data exploratory technique. The PCA score-plot shows two unsupervised groupings stimulated us to continue the multivariate analysis with OPLS-DA, as described by reference 32.

Round 2

Reviewer 1 Report

Comments and Suggestions for Authors

The authors addressed all my concerns. Only a minor comment, at line 326, I think "figure 1" should be changed with "figure 4".

In my opinion the manuscript is now suitable for publication.

Author Response

Sir, please correct the typing errors concerning the Author "Nicola Culeddu" (reported as Culeeddu) and the e-mail of the author "gbartolomeo@unime.it" (gbatolomeo@unime.it)

A.1

  1. Page 4, lines 150-157. According to our opinion the few discrepancies are reminiscent of possible interferences in the Folin-CiocaÌ‚lteu measurements [28]. We would like to mention that the NMR method quantifies a fifth elenolic derivative called Elenolide [29] which is not considered as an electron donating com-pound and it is not considered in this comparison. It should be noted that samples N_7, N_8 and N_12 with remarkable presence of TPC are also those displaying the higher deviation with an underestimate of the spectrophotometric method”.

The authors are not aware of a publication with a detailed discussion and comparison of the NMR total phenolics method with the FC spectrophotometric method which has several disadvantages (Anal. Chim. Acta 2011, 688, 54-60). A proper reference and discussion should be provided.

R.1

Thanks to the appropriate suggestion we have changed the paragraph doing our best in the comments on this subject reported below, on the other hand, we have to notice that the mentioned studies refer to methanol extracts.

“It looks shared that the popular, easy, and direct spectrophotometric method is affected by the use of a three-functional hydroxy group chemical reference (gallic acid) and by the interferences of inhibitory, additive and enhancing type [28]. Provided that the general trend is pretty respected the few discrepancies are reasonably due to the different number of functional groups per molecules and to the possible interferences in the Folin-Ciocâlteu measurements [29].”

A.2

  1. Page 9, lines 338-343. Sample preparation was developed after several test experiments. The experimental setup is our best optimization to keep a reasonable field homogeneity affecting line-with and resolution, sensitivity, and chemical stability. Just for instance, we would like to point out that many studies suggested to add small amounts of d6-DMSO to decrease the sample viscosity and therefore the field homogeneity, however our experiments demonstrate that this procedure chemically affects minor components as phenols, preventing the suitable detection and quantification.”.

This part of a paper should be completely revised. DMSO-d6 does not chemically affect minor components as phenols preventing the suitable detection and quantification. On the contrary, the strong hydrogen bonding interaction of DMOS-d6 with the phenol OH group reduces significantly the proton exchange processes, thus, resulting in very sharp OH resonances. This permits accurate detection and quantification. The method is not limited to phenol OH groups but can be applied to any functional group with labile protons such as carboxylic groups of free fatty acids. Some relevant references are the following: Chem Commun. 2010, 46, 3589- 3591; Lipid Technol. 2012, 24, 279-281; Phytochem. Anal. 2017, 28, 159-170. Of particular interest are the review articles: J. Pharm. Biomed. Anal. 2014, 93, 43-50; Molecules 2014, 19, 13643-13682 and Process 2020, 8, 410. Figure 3 (see below) demonstrates that the DMSO-d6 method is, perhaps, the only direct method which can differentiate free fatty acids from oxidized hydroperoxide derivatives due to the significant chemical shift difference and, thus, excellent resolution of COOH and OOH groups in the highly deshielded region. Accurate integration, therefore, can be achieved without interference of other functional groups (e.g., aromatic and aldehyde resonances).

R.2

This interesting suggestion prompted us to better argue the solvent choice. The very important mentioned refererences – which have been added in the text -  were held into consideration keeping in mind that: a) our quantification protocol, based on the CHO group in CDCl3, is not affected by the hydrogen bonding network generated by the chosen solvent; 2) in the specific case of secoiridoids, any polar solvent is risky as it changes the profile of such components (Food analytical Methods https://doi.org/10.1007/s12161-019-01508-5). In order to demonstrate this point, we have added the figure 4A.

Thankfully we changed

“Just for instance, we would like to point out that many studies suggested to add small amounts of d6-DMSO to decrease the sample viscosity and therefore the field homogeneity, and this is supported by several quantification studies in polar media [37,38] as well as in CDCl3 medium (for vegetable oils or extracts) [39,40]. On the other hand our experiments demonstrate that, in the case of secoiridoid derivatives, polar media trigger isomerization and decarboxylation processes preventing the specific quantification as shown in the figure A4. We point out that, unlike the mentioned studies this quantification is based on the CHO aldehydic group whose signal is not affected by the potential hydrogen bonds of the polar media”

A.3

  1. Page 10, lines 407-409. Calibration of experiment A was basically performed on the methyl group of the β-sitosterol signal to (δH = 0.738 ppm) provided that, when TMS was in the sample, its signal was always δΗ=0.0± 0.005 ppm.”.

TMS is volatile, therefore, is not appropriate for integration and, thus, quantitative results.

R. 3

We thank for spotting that the sentence was not clear. Of course, TMS is not a quantitative reference and is not used for this purpose (as explained in the following paragraph the relative intensity of fatty acid signals through the MARA-NMR procedure are used for the quantitative studies), it is rather used just in few samples to double check the good internal frequency  reference (not quantitative!!) of b-sitosterol.

“The experiment A was frequency calibrated by using the internal b-sitosterol signal (δH = 0.738 ppm). The frequency of this signal is double checked in several samples by using the external frequency of TMS (δΗ=0.0) which never shifted the spectral profiles more than 0.005 ppm”

R.4

  1. Figure A2.

Figure A2 is very confusing due to overlapping of several spectra some of which are very noisy. The authors should present the various spectra in a consecutive way and not overlapped.

A.4

We have modified the figure, actually the stack-plot does not look organized or organizable combining also the assignment, so thank to the suggestion we have chosen just one representative experiment to evidence assignments

  1. Instead of the term “innovative NMR method”, a more moderate term could be used.

A.5

We have replaced the part with “novel NMR method” hoping it is appropriate